# Self-Abrading Servo Electrode Helmet for Electrical Impedance Tomography

**DOI:** 10.3390/s20247058

**Published:** 2020-12-09

**Authors:** James Avery, Brett Packham, Hwan Koo, Ben Hanson, David Holder

**Affiliations:** 1Department of Surgery and Cancer, Imperial College London, London SW7 2AZ, UK; 2Department Medical Physics and Biomedical Engineering, University College London, London WC1E 6BT, UK; brett.packham.09@ucl.ac.uk (B.P.); hwan.koo.09@ucl.ac.uk (H.K.); d.holder@ucl.ac.uk (D.H.); 3Department of Mechanical Engineering, University College London, London WC1E 7JE, UK; b.hanson@ucl.ac.uk

**Keywords:** electrical impedance tomography, stroke, brain imaging, bioimpedance electrodes

## Abstract

Electrical Impedance Tomography (EIT) is a medical imaging technique which has the potential to reduce time to treatment in acute stroke by rapidly differentiating between ischaemic and haemorrhagic stroke. The potential of these methods has been demonstrated in simulation and phantoms, it has not yet successfully translated to clinical studies, due to high sensitivity to errors in scalp electrode mislocation and poor electrode-skin contact. To overcome these limitations, a novel electrode helmet was designed, bearing 32 independently controlled self-abrading electrodes. The contact impedance was reduced through rotation on an abrasive electrode on the scalp using a combined impedance, rotation and position feedback loop. Potentiometers within each unit measure the electrode tip displacement within 0.1 mm from the rigid helmet body. Characterisation experiments on a large-scale test rig demonstrated that approximately 20 kPa applied pressure and 5 rotations was necessary to achieve the target 5 kΩ contact impedance at 20 Hz. This performance was then replicated in a simplified self-contained unit where spring loaded electrodes are rotated by servo motors. Finally, a 32-channel helmet and controller which sequentially minimised contact impedance and simultaneously located each electrode was built which reduced the electrode application and localisation time to less than five minutes. The results demonstrated the potential of this approach to rapidly apply electrodes in an acute setting, removing a significant barrier for imaging acute stroke with EIT.

## 1. Introduction

### 1.1. Background

#### 1.1.1. Acute Stroke

Stroke is a leading cause of morbidity and disability in industrialised nations [1]. It is the third leading cause of lost life years [2] and estimated that total aggregate annual costs of stroke to the UK is £25.6 billion [3]. Stroke has two main causes: ischaemia, where bloody supply is interrupted due to a blood clot, and haemorrhage due to a vascular rupture. Earlier treatment through thrombolytics results in better patient outcomes with the benefits limited to the first 3 to 6 h after onset [4]. Thrombolytic agents can only be used to treat ischaemic strokes as they are potentially fatal for patients with haemorrhages, so it is vital that cause of stroke is identified prior to treatment. Currently this cannot be achieved without neuroimaging, and in the UK, the National Institute for Health and Clinical Excellence (NICE) guidelines stipulates that brain imaging should be performed “immediately (within the hour)” for patients with a suspected stroke [5]. However, studies have shown that transport to CT or MRI scanners and completion and reporting of the scan take at least an hour [6], and that the majority of stroke patients are arriving at emergency departments 3 h after onset [7,8]. It is clear therefore that faster imaging and differentiation of acute stroke is vital to prevent patients being denied essential treatment during this timeframe.

#### 1.1.2. Electrical Impedance Tomography of Acute Stroke

Electrical Impedance Tomography (EIT) is a tomographic medical imaging technique which produces the internal electrical conductivity of a subject from multiple impedance measurements using surface EEG electrodes [9]. EIT has been proposed as a solution to enable early neuroimaging in ambulances or diagnostic centres without other scanning systems [10,11]. Ischaemic and haemorrhagic strokes produce different changes in conductivity spectra relative to healthy brain, and this contrast enables EIT imaging and subsequent differentiation. Cell swelling in ischaemia causes a reduction in extracellular space, resulting in a conductivity decrease at low frequencies (<100 Hz) [12]. Whereas a haemorrhage produces a high conductivity region due to increased local blood volume [11,13]. Being inexpensive, portable and sensitive to tissue conductivity changes resulting from stroke, EIT has the potential to provide diagnosis within an ambulance or primary care service and enable time critical treatment.

Imaging stroke in an acute setting presents several practical and numerical challenges. Most EIT applications produce images of changes in conductivity after subtracting a baseline or reference data frame, known as “Time Difference” EIT [14]. However, this is not possible in stroke diagnosis as there is no “pre-stroke” data available as a reference. Therefore frequency difference EIT algorithms have been developed which use data collected at multiple frequencies at a single time point [15,16]. However these methods are not able to replicate the tolerance of Time Difference methods to modelling errors in electrode position and anatomy [17,18]. Algorithms have been developed which to compensate for these errors [19,20,21], but these still have a low error threshold of approximately 1 mm.

Currently, patient specific Finite Element Models (FEMs) are generated from CT or MRI scans are used to minimise errors from inaccurate geometry [15,22]. However, this is not possible when performing EIT in an ambulance. While studies have shown that a sufficiently detailed generic mesh may be adequate for time difference [23] or machine learning [24], a frequency difference method robust to these errors has not yet been developed. Therefore successful imaging will require co-registration and deformation of existing meshes [25].

#### 1.1.3. EIT Data Collection and Electrode Placement

The current procedure for applying electrode for brain EIT studies, Figure 1, closely follows that used for EEG recordings [10,26,27,28]. Ag/AgCl electrodes are placed on the scalp of the patient by hand after preparation of the electrode sites with alcohol and abrasive paste to reduce the impedance of the electrode-skin interface known as contact impedance. Manual application of the electrodes in this manner is labour intensive and impractical in an acute setting. EEG caps or headnets can facilitate the placement of electrodes to reduce the preparation time but still require manual abrasion to be performed [28,29]. Head nets however offer limited benefit in terms of localising electrodes with mismatches above 10 mm recorded in the literature [30,31]. Thus, localisation through photogrammetry or electromagnetic methods are necessary, which have errors in the 1–2 mm range [30,32,33].

EIT also has more stringent requirements for acceptable contact impedances than EEG, e.g., 1–3 kΩ at 1 kHz [28], compared to 10–20 kΩ for commercial EEG systems. While EIT algorithms are robust to a wide range of contact impedances [15,21], high contact impedance still results in adverse errors due to instrumentation nonidealities. This is primarily due to the current injection, which is more sensitive to contact impedance mismatch than voltage recording due to comparatively lower output impedances of the order of 1 MΩ [34,35]. The need to inject current through the electrode precludes the use of active electrodes common in EEG systems without bespoke circuits [36]. This requirement also prevents the use of “dry” electrodes which do not require abrasion or electrolyte paste and have been shown to perform comparably to wet electrodes in EEG, despite higher contact impedances [37].

Therefore, it remains essential to perform abrasion of the electrode contact site for EIT measurements. Self-abrading electrodes where the abrasion is performed in-situ by an electrode with an abrasive surface have been proposed as a solution to rapidly reduce the contact impedance for breast EIT [38]. Holding the tissue in place pneumatically while the electrode was rotated, it was possible to reduce the contact impedance to similar values achieved with traditional abrasion and Ag-AgCl electrodes, 5 kΩ at 20 Hz. This system is not suitable for stroke EIT as it requires significant intervention on the part of the clinician and do not offer electrode localisation.

### 1.2. Rationale—Self-Abrading Electrode Helmet

Successful EIT imaging of stroke from scalp electrodes in an acute setting requires an improved electrode application method which correctly locates electrodes in known positions with low contact impedance. The solution proposed in this work is to automate the electrode application process using a novel electrode helmet bearing a multitude of servo-controlled self-abrading electrodes. The design extends existing self-abrading electrodes by incorporating position sensors and automating the abrasion using a contact impedance feedback loop. Control of the contact impedance with a servo mechanism not only allows the contact impedance to be minimised upon initial application, but also maintained throughout recordings. To limit misalignment with the scalp of the subject, the direction the electrodes move during compression is based on the surface normal vectors of a reference head model.

Thus, the primary goal for the design of the helmet was to place the electrodes on the scalp in the correct positions and reduce the contact impedance to match conventional techniques [10,28] and previous self-abrading electrodes [38]. Specifically, <5 kΩ below 100 Hz within 10 s per electrode. The secondary application for the helmet to reduce the changes in contact impedance over time, by monitoring and repeating abrasion to reduce measurement drift which has been identified as a significant barrier in brain monitoring applications [26,39].

#### Ambulance and Patient Considerations

To design a system to apply electrodes to the scalp of a patient in an ambulance that is robust, consideration must be given to the environment and pateint safety. To reduce patient discomfort, the amount of abrasion performed should be minimised and the device should be back-drivable, i.e., be able to adjust to patient movement during recording. The weight of the helmet should be minimised, and the centre of mass should also be aligned with the neck to prevent uncomfortable straining. It is essential that the electrodes be positioned rapidly as the time required to collect a full spectrum EIT protocol [28] is similar to the mean ambulance journey time for stroke patients in the UK [40].

### 1.3. Experimental Design

In clinical studies the abrasion process is repeated until a satisfactory contact impedance is reached [10,28], and investigations into the impedance of skin layers do not quantify the mechanical forces [41]. Thus, the first step was to quantify the effects of variations in force and torque on abrasion effectiveness in vitro, through measurements on conventional manual abrasion and a large-scale prototype. Subsequently, a miniaturised self-abrading electrode unit was designed to meet these specifications. Finally, a 32-channel electrode helmet and controller were designed and tested in subject scalp recordings.

## 2. Characterisation of Abrasion

### 2.1. Test Rig Components and Validation

A prototype single electrode device, Figure 2, was designed and constructed with two independently controlled degrees of freedom: linear normal to the skin surface with closed loop control of force, and rotation about this axis with closed loop speed control. The electrode was connected to a spindle aligned normal to the test object surface and the impedance was measured continuously during abrasion between the 10 mm brass actuated electrode, and a 50 × 70 mm silver reference electrode behind the orange skin test object. The whole system was controlled through LabView using a NI 6221 DAQ, with a control loop speed of 15 ms or 66.67 Hz. To simulate the impedance and mechanical properties of human skin, test objects were created from sheets of navel orange skin (Citrus Sinensis). The suitability of which was tested in experiments comparing the impedance spectra of human skin before and after abrasion.

#### 2.1.1. Force Control

The force applied by the electrode onto the test object during abrasion was controlled via linear actuator with attached spring and a load cell placed behind the object under test, Figure 3a. The linear displacement is converted to an applied force through a 2 kNm^−1^ spring to provide a maximum force of 20 N at the maximum 10 mm travel. A simple on-off closed loop feedback scheme was used with a deadband of 0.2 N to reduce limit cycling. The RMS error of the force applied across all experiments was 0.228 ± 0.058 N.

#### 2.1.2. Rotation Control

Rotation of the electrode was achieved through a DC motor with custom rotary encoder and a transmission system comprising of a gear train and a keyed joint, Figure 3b, which allowed the spindle to translate axially during rotation. A target rotation speed of 60 RPM was used, based on a comfortable speed for self-abrading electrodes in the literature [38]. Through control of the PWM duty cycle, the rotation could be set to a range of 10–60 RPM with 0.1 to 0.4 Nm stall torque.

#### 2.1.3. Impedance Measurement

An impedance measuring circuit was implanted using the USB DAQ to measure the voltage drop across a known resistor and the load, which was connected to the 10 mm brass self-abrading electrode and the 50 × 70 mm silver reference behind the test object. Thus, the impedance measured is inclusive of both contact and electrode impedances. The impedance circuit, Figure 4, uses the USB DAQ to generate a constant voltage a specified frequency, and the differential voltages across $R_{known}$ and $Z_{load}$ were amplified and recorded by the DAQ. Using the known value of 10 kΩ for $R_{known}$ the impedance could be calculated using the following equation:(1)$$Z_{load}=\frac{V_{load}}{I}=\frac{V_{load}R_{known}}{V_{C}}$$

A carrier frequency of 20 Hz was chosen as lower frequencies represent the most stringent test as skin impedance is largest and was the lowest frequency used in previous stroke EIT studies [10]. The impedance error was less than 0.25% from 50 Ω to 100 kΩ at 20 Hz as compared to HP 4284A Impedance Analyser. A parallel resistor capacitor circuit based on the equivalent circuit for the stratum corneum [41] was created to test the accuracy of the impedance measurement circuit across frequency. The output sample rate of the DAQ limited the range below 10 kHz, where the error was found to be 0.3% or less. The active electrode was a 10 mm brass disc with a chamfered pyramidal pattern machined into the surface, which was constrained in the radial direction during abrasion by a retaining bush. Silver epoxy connected the electrode to the spring to provide an electrical path through the spindle and slip ring to the DAQ. A total electrode impedance of 450 Ω at 20 Hz was measured in contact with a saline tube using the HP 4284A.

### 2.2. Quantification of Conventional Abrasion and Target Impedance

To quantify the force applied to the skin surface during manual abrasion and verify the orange skin test objects, experiments were performed replicating conventional electrode application, Figure 5a. Three test samples were abraded using the methodology used in previous studies with EEG electrodes [10,28]. The surface was abraded using NuPrep gel and an applicator with a surface area of approximately 25 mm^2^ in contact with the tissue, and the contact impedance decrease was measured between two silver reference electrodes.

The average root-mean-square force applied normal to the sample surface, Figure 5b was 0.62 N ± 0.09, which equates to a pressure of 24.8 kPa over the 25 mm^2^ surface area. The impedance across frequency before and after abrasion, Figure 5c, fell from approximately 30 kΩ to 4.5 kΩ at 10 Hz and the spectra decreased monotonically to 5 kΩ and 800 Ω respectively. The orange skin spectra demonstrate the same broad trend as those observed in human skin by Yamamoto and Yamamoto [41] with 1 kΩ values approximating the spectra observed after 9 strippings or removal of 72 μm of the stratum corneum. This is greater than the expected abrasion depth of 120 μm in human skin, and resulting impedance of approximately 400 Ω. Thus, the orange peel test objects were found to provide realistic impedance changes, albeit with higher final values. Given the electrode impedance of 430 Ω and the greater impedance of orange skin compared to human skin after abrasion, a target of 6 kΩ impedance was set for subsequent tests.

### 2.3. Automatic Abrasion

The decrease in contact impedance by abrasion through rotation of the electrode was measured with the speed and force control active. First the experiment was performed at full speed and torque with 3 samples for force levels of 2, 3, 5, 7.5 and 10 N. The mean impedance at 20 Hz and rate of decrease per sixth rotation was calculated across all samples.

Figure 6a demonstrates that increasing force decreased both the initial and final impedances obtained after abrasion. In all cases the impedance decreased monotonically with electrode rotation with the greatest change observed in the first 2 full rotations. This is reflected in the rate of impedance decrease Figure 6b, which was predominantly unaffected by increasing force, with the exception of the first half rotation, where the lowest forces demonstrated the greatest change as a result of the significantly higher initial impedance. All forces applied above 5 N reached the modified target threshold of 6 kΩ.

To understand the effects of decreasing rotational torque, the experiments were repeated at a single 5 N applied force, with the DC motor torque controlled by the PWM duty cycle. Below 0.25 Nm, it the torque was not sufficient to overcome the friction forces at the electrode surfaces and the electrode did not rotate. Above this threshold, the impedance decrease per rotation, Figure 6c did not significantly decrease, and the threshold impedance was reached within 5 full rotations and 10 s. The threshold torque necessary to abrade increased linearly from 0.2 Nm at 2 N contact force to 0.45 Nm at 10 N, demonstrating an approximately constant coefficient of friction μ.

The impedance spectra measured with two reference electrodes, before and after abrasion, Figure 6d, are in good agreement with the results of manual abrasion with an approximately constant value of 1 kΩ between 1 Hz and 1 kHz, which suggests a removal of similar amount of the outer skin layer.

### 2.4. Proof of Principle on Human Skin

Finally, to assess whether the performance translated from the test rig to a more clinically representative scenario, automated abrasion was performed on the forearm of a human subject. The forearm was chosen as it has a similar a thickness of stratum corneum to the scalp [42]. A 50 × 70 mm silver reference electrode was placed close to the elbow after preparation using conductive abrasive paste. As it was not possible to use the load cell in this case, forearm was positioned such that the spring was displaced to the same at position which provided 7.5 N applied force previously. The experiment was repeated three times with a “dry” electrode and then with abrasive paste applied.

Using abrasive paste the contact impedance decreased from 10 kΩ to 800 Ω in less than 2 rotations, whereas without paste the decrease was from >50 kΩ to 5 kΩ in 9 rotations, Figure 7a. The speed of rotation was 0.22 Hz or 13.2 RPM for dry abrasion, and 0.7 Hz or 42 RPM for wet abrasion with paste due to the reduced coefficient of friction. Without paste the impedance decreased in a similar manner to that observed in orange skin, with the majority of the impedance decrease occurring the first 3 rotations and the target 5 kΩ reached in 9. The impedance spectra Figure 7c show the contact impedance was decreased across the frequency range.

The addition of abrasive paste greatly reduced the amount of abrasion required. After only one entire rotation the impedance was reduced well below the 6 kΩ threshold impedance to a value of 800 Ω. The abrasive paste significantly reduced the contact impedance before any abrasion took place, from 30 kΩ to 10 kΩ at 20 Hz and facilitated the abrasion from the electrode, clearly evident in the comparison Figure 7a.

As with the manual abrasion characterisation Figure 5, the impedances measured were corrected for electrode impedances and compared to those from Yamamoto and Yamamoto [41], Figure 7c. The results suggest that without paste the impedance has decreased greater than the equivalent of 6 tape strippings or the removal of 48 μm of the Stratum Corneum, whereas abrasion with paste improved upon manual abrasion with impedances lower than observed after 9 strippings or 72 μm removed.

### 2.5. Implications for Single Self-Abrading Unit

The large-scale prototype demonstrated the principle of skin abrasion with impedance feedback can meet the requirements to lower the contact impedance within a target 6 kΩ and 10 s. Based on these results it is clear that the number of electrode rotations has the greatest effect on the impedance decrease, rather than the rotation speed or torque applied. Therefore, a position-controlled actuator is most suited for miniaturisation, as the position and thus number of rotations can be controlled directly. The transmission system can also be significantly simplified as the higher speeds and torques did not significantly improve the rate of impedance decrease. The results from the experiments suggest the abrasion efficiency is independent of the applied force above a certain threshold and thus the force does not require a separate control. A force between 5 N and 10 N was sufficient to reach the target impedance within 10 rotations, and the forearm experiments demonstrated that the impedance could be controlled without the force control loop active. Such a wide range of acceptable forces does not fully justify the added complexity of a force control loop. Reducing the electrode diameter will both allow for easier access to the scalp through hair and decrease the contact surface area resulting in lower friction forces and thus torque requirements to overcome them.

## 3. Self-Abrading Electrode Unit

### 3.1. Single Unit Design

Based on the conclusions drawn from the abrasion quantification experiments, a self-abrading servo electrode unit, Figure 8, was designed, with an electromechanical rotary actuator and passive force provision. Abrasion was achieved through a servo motor with a rotation range of 180${}^{\circ }$ oscillating around the central position, with a plain bearing to maintain concentricity. This negated the need for a slip ring for the wiring to the electrode and position sensor. Force was applied to the scalp through the spring inside the linear potentiometer. Abrasive paste was attached to each electrode prior to use, to reduce the initial contact impedance, improve abrasion efficiency, and reduce the coefficient of friction. The compression of the spring also translated the contact wiper inside the potentiometer $X_{P}$, from which it was possible to obtain the position of the electrode along the axis of rotation $X_{E}$. The electrodes were then placed in a rigid helmet where the reference positions $X_{S}$ were known with high precision *a priori*. Therefore, when placed on the scalp the positions of the electrode tips could be calculated from the single potentiometer measurement. Each of the units were self-contained individual 3D printed casings, which enabled simple replacement or transfer of the units. An Arduino Due was used to control the motors, sensors and impedance measurements, and communicate to a PC through Serial interface.

#### 3.1.1. Motor

The servos used in the design had a a measured stall torque between 0.203–0.234 Nm, which is less than the 0.3–0.4 Nm minimum torque required to abrade in the previous section. However, the friction expected at the electrode interface for the miniature design is reduced, as the area of the electrode is less than half, and the inclusion of paste significantly reduces the coefficient of friction μ. To reduce potentially uncomfortable sudden changes in acceleration, “S Curve” velocity profiles were used when to change the direction of rotation [43].

#### 3.1.2. Position Sensor and Passive Force Actuation

Position feedback in the electrode unit was achieved through a 9605 spring return linear position sensor from BEI Sensors. Movement of the actuator compresses the internal spring and adjusts the position of the wiper on the resistive strip, thus the voltage across the wiper gives a measure of the distance travelled, from which the force can also be calculated using the spring constant *k*. A characterisation experiment was performed using the test rig in the previous section and a laser displacement sensor (LDS) (optoNCDT 1607, micro-epsilon, UK), Figure 9. The 9605 was attached to the spindle of the stepper motor and advanced onto the load cell while voltages were recorded from the LDS, potentiometer and load cell simultaneously.

The potentiometer had a maximum repeatability error of 0.78% across the complete 13 mm travel, equivalent to approximately 0.1 mm. Some fluctuations in the applied force were observed, Figure 9b, resulting from the stiction of the potentiometer wiper. The spring was preloaded with a force of 0.5 N measured at 0 mm travel. The maximum force applied at full compression was 4.6 N, which for an electrode of 10 mm diameter equates to a pressure of 28.6 kPa, close to the maximum used in the abrasion characterisation tests. Thus, adequate force is applied for ideal abrasion for over half of the travel of the electrode.

#### 3.1.3. Electrode and Impedance Measurement

A prototype miniaturised electrode with a pyramidal pattern was found to cause discomfort after only a few minutes after abrasion during preliminary testing with the higher applied forces. Therefore, smooth electrodes were also created and tested using abrasive paste in comparison to the pyramidal abrasive pattern. The electrodes were constructed from 316 stainless steel, with 10 mm diameter and a 0.5 mm chamfer on the edge in contact with the subject. The electrode was wired to the controller via the spring within the potentiometer, Figure 10a.

The impedance measurement circuit used in the previous section was adapted for use with the Arduino Due, Figure 10b. A 1 V amplitude sine wave was generated using the on-board DAC with a 1.65 V DC offset which was removed with a DC blocking capacitor C1 before the electrodes. A bias voltage was then added to the voltages $V_{known}$ and $V_{load}$ at the input to the Arduino ADCs. MOSFETs were used to discharge all capacitors to ground between measurements. The voltages recorded were zero phase filtered with a 3rd order Butterworth band pass filter and the amplitude determined by the root mean square of the result. Finally, the impedance was again calculated from the ratio of the in Equation (Equation 1). The accuracy of the impedance measurement circuit was tested by comparing impedances measured on 10 to 1 MΩ resistors, with 10 repetitions, to that measured by the HP 4284A. The error was less than 1% from loads between 100 and 50 kΩ, Figure 10c, the maximum noise recorded within this range was a standard deviation of 1.2%, Figure 10d. Thus, the performance of this simplified circuit is similar to that of the impedance measurement circuit used in the test rig, despite the fewer averages and lower resolution ADC.

The impedance control loop was a simple on/off controller with a deadband of 2% to prevent limit cycling. The initial target impedance was 5 kΩ. If this impedance was not achieved within 10 rotations, the abrasion was stopped, and the target impedance for subsequent iterations was set to the lowest impedance measured during initial abrasion.

#### 3.1.4. Unit Construction

Construction of the electrode unit required modification of the potentiometer and printing of custom components. The tip of the spring potentiometer actuator was drilled to allow connection of the electrode which was held in place by an adaptor. Three wires were soldered onto the contacts to the potentiometer and passed out of the body of the unit using the same path as the electrode wiring, Figure 10a. The electrode was then placed within the adaptor using an interference fit. The potentiometer was then connected to the spindle of the servo motor via a 3D printed adaptor and the complete unit placed in an individual housing. The mass of the complete electrode unit inclusive of all 3D printed adaptors and housings was 31.4 g.

### 3.2. Single Unit Experiment

The experiment on the human forearm was repeated with the newly designed single electrode unit. The electrode was compressed onto the skin at 75% of the maximum travel, equivalent to 3.45 N or 21.4 KPa. Abrasion was performed with a plain electrode and a pyramidal patterned electrode, Figure 11a, both with abrasive NuPrep paste.

The abrasion was similarly effective with both a pyramidal and smoothed electrodes, Figure 11, achieving a contact impedance less than 8 kΩ after one rotation. After two rotations the smoothed electrode reached a stable 7.5 kΩ, whereas the pyramidal electrode continued to decrease the contact impedance to 3.2 kΩ after 8 rotations. The results demonstrated that the miniaturised unit was able to replicate the abrasion results observed in the experiments in the previous section. Also, importantly for subsequent designs, the impedance could be reduced sufficiently, without the added complexity and discomfort of the pyramidal electrode surface.

### 3.3. Four Channel System

To further validate the design of the electrode units, the abrasion was performed using a linear array of four electrode units, Figure 12a, positioned on the shin aligned with the tibia. The array was held in place by an elastic bandage which was tightened such that each of the four electrodes reported ≈ 85% compression, and an EEG ground electrode was positioned on the calf. Abrasion was performed with the application of NuPrep abrasive paste, and also with the addition of a layer of EleFix conductive gel on two electrodes. The experimented was repeated three times for both cases on the same subject on different days, with the EleFix gel assigned randomly to a pair of electrodes, and the results presented as a mean and standard deviation.

A long-term recording was also collected on the shin to directly compare the measurement drift of the actuated electrodes with conventional EEG cup and paste electrodes. The two channels on the actuated system were replaced with two EEG electrodes which were manually abraded with NuPrep paste and EleFix conductive gel was added for a single channel. Measurements were collected over the course of half an hour, with the impedance control loop reactivated every 60 s. The abrasion was only reactivated if the measured impedance value was greater than 1% of the previous value.

The four-channel system reduced the contact impedance rapidly using the NuPrep paste only, decreasing the impedance by 25 kΩ to 6.7 kΩ with one half rotation, Figure 12b. Only a further reduction of 1.3 kΩ was achieved with the remaining four complete rotations, with a minimum of 5.4 kΩ. The addition of a layer of conductive EleFix paste between the electrode and NuPrep abrasive reduced the initial impedance by 12 kΩ but reduced the efficiency of the abrasion, achieving a minimum impedance of 8.1 kΩ.

During the half hour recording the abrasion was continuously reactivated for the actuated electrode with NuPrep only, Figure 12c, whereas it was not repeated beyond the initial application with the addition of the EleFix. In each instance, only an additional half rotation was required to reduce the contact impedance to its previous minimum. The impedance of both the actuated electrodes decreased over the course of the experiment, despite no additional abrasion in the case of the EleFix and Nuprep electrode. This is likely due to the pressure applied by the spring, which was not applied to the conventional EEG electrodes. The impedance of EEG electrode with NuPrep linearly increased over the course of the recording from 3.91 kΩ to 4.35 kΩ. The EEG electrode with the conventional combination of NuPrep and EleFix was more stable over time, with an increase of only 100 Ω from the initial 2.3 kΩ. These long-term recordings demonstrated the ability of the unit to correct drifts in impedance when using NuPrep paste only. The drifts in contact impedance were not visible in the electrode with a layer of EleFix paste and is a function of the suboptimal electrochemical interaction of the electrode-gel-skin interface when using stainless steel. This problem should therefore be overcome in future designs implementing Ag/AgCl electrode surfaces.

## 4. Electrode Bearing Helmet

### 4.1. Helmet Design

The basis of the helmet design was a high quality, smoothed CAD model, which was first scaled such that the length and breadth matched the average size as found in anthropometry literature, 189 and 146 mm, respectively. The resultant circumference was 549 mm, which underestimated the average circumference as found by [44], likely a result of the smoothing and lack of hair in the CAD model. The 32 electrodes were located by replicating the EEG 10–20 placement strategy on the solid model as described by [45], which provided an even coverage of the scalp and adequate spacing for all electrode units.

The reference model was trimmed below the inion-nasion plane, and the resultant surface was offset by 10 mm to provide the inner surface of the helmet, a further 10 mm offset was used to create the outer surface. This offset combined with a 13 mm travel covers over a standard deviation of the head sizes described by [44]. Vectors normal to the scalp surface at each electrode point were created as references axes for alignment of electrode housings Figure 13. The electrode units were angled to reduce support material and subsequent post processing. Centre of Mass analysis was performed on the CAD model to ensure the it was aligned to within 5 mm of the coronal-sagittal axis. The mass of the completed helmet frame was 604 g. Finally, markers were added along the inion-nasion line and the auricular points, to aid in correct helmet alignment

### 4.2. Controller

An Arduino Due was used a central controller for all 32 channels, consisting of smaller sub-controllers for the servo motors and potentiometer measurements, and a single multiplexed impedance measurement circuit, Figure 14. The Arduino controller was battery powered, isolated via wireless serial communication, and a LabView front end was designed for data logging and visualisation. The servos were supplied with 6 V directly from the battery, controlled through two daisy chained Adafruit 815 PWM/Servo Drivers and switched off when not in use. On the same I2C bus eight Adafruit ADS1015 12-Bit ADC boards were used to measure the voltage across each potentiometer. Initial position measurements were averaged over 1000 samples for each electrode. During the “idle” state, the positions were re-sampled every second, and the user was warned if any large changes were detected. The 37 channel switch network designed for the UCL Scousetom EIT system [46], was used to address each of the electrodes on the helmet. Impedance measurements were collected with respect to an adhesive Ag/AgCl reference electrode placed on the forehead. This was chosen to minimise the contribution to the contact impedance measured and provide a stable reference over time so as not to mask the behaviour of the electrodes under test.

### 4.3. In Vivo Experiments

Finally, the complete 32 channel helmet was tested on the scalp of a volunteer. Abrasive NuPrep paste was applied to each electrode and the helmet positioned on the scalp of the subject and the alignment verified by the inion-nasion and auricular markers. The abrasion routine was activated for each of the 32 electrodes in turn. The resulting contact impedance decrease was expressed as a mean and standard deviation across all channels. Finally, the minimum impedance during the initial abrasion as a function of electrode position was calculated and averaged into 0.5 mm bins.

The contact impedance was successfully reduced through automated abrasion on all 32 electrodes, within five complete rotations or nine seconds. The entire serial abrasion routine was completed in 4.6 min. In all cases, the largest decreases occurred within the first complete rotation, and comparatively little contact impedance decreases in subsequent rotations. In general, the impedance was reduced to approximately 7 kΩ or less for all electrodes, Figure 15. There was, however, considerably larger variation between channels than that observed in the measurements on the leg, Figure 12. The minimum impedance achieved was strongly dependent upon the compression of the spring Figure 15, in line with the results in Section 2. Abrasions at locations where the spring was compressed by 2 mm or less, were significantly less effective. Beyond 8 mm, or approximately half the potential force applied, there was little improvement in the minimum impedance achieved.

## 5. Discussion

### 5.1. Self-Abrading Electrode

The design of the electrode unit based on the abrasion quantification experiments demonstrated similar performance despite considerable simplification. The electrode unit reduced the contact impedance to less than 5 kΩ in less than two rotations with a patterned electrode, matching the performance of previous self-abrading electrode designs [38]. The simplified plain electrode demonstrated the principle but due to the electrode material choice achieved a final impedance just above the target value.

The four channel experiments further demonstrated the impedance reduction achieved through the miniaturised electrode unit, while simultaneously offering a more stringent test of the controller. Experiments demonstrated a repeatable 25 kΩ decrease in impedance within one half rotation, and a minimum of 5.4 kΩ within five rotations. The results of the long-term recordings were less conclusive, as drifts up to 5 kΩ were observed. This effect was not visible in the electrode with a layer of EleFix paste and is therefore likely a function of the electrochemical interaction of the electrode-gel-skin interface, and errors in the impedance measurement. The contact impedance of all 32 electrodes within the helmet were automatically reduced to approximately 8 kΩ within five minutes on the scalp. Below 8 mm travel, the performance of the electrode units was proportional to the compression of the internal spring. In locations where the compression was greater than 8 mm, there was no significant difference in minimum impedance. This correlates with the results from the test rig, where beyond a certain pressure the abrasion was not improved.

### 5.2. Assessment of Helmet Design

For the primary application of imaging acute stroke in an acute setting, the results obtained represent a clear proof of principle, if not establishing a methodology for recording EIT in an ambulance. The helmet met the requirements for reducing the contact impedance on 32 electrodes automatically within five minutes. The mean minimum impedance, 8.2 kΩat 20 Hz, was higher than the target 5 k, as a result of the electrode material used in the prototype. The results in Figure 12 demonstrate that creating custom Ag/AgCl electrodes would reduce the minimum impedance below the target and reduce measurement drift.

A theoretical electrode localisation accuracy of 0.1 mm was achieved through the use of spring potentiometers, a potential improvement upon the localisation achieved through photogrammetry. However this accuracy is reduced by the manufacturing tolerance, approximately 0.2 mm using the same hobbyist FDM 3D printer [45], and alignment of the helmet on the head. Using the helmet precludes direct optical measurements of the electrode locations [47], however a hybrid system where the helmet location and orientation is optically tracked and then combined with the accurate potentiometer readings could result in less error than direct optical measurements. Further experiments are necessary to assess the accuracy of positioning the electrodes in the target locations on varying subject anatomy. While EIT algorithms have been shown to be sensitive to random electrode localisation errors of approximately 1 mm [15,19], the sensitivity to errors arising from misalignment of the helmet, which can be approximated as rotations around each anatomical axis, have yet to be tested. The BFSD-EIT algorithm has been shown to be robust to errors of up to 30 mm if symmetry is maintained [21], which may be the case for rotations of the helmet around the lateral axis.

The helmet was designed to cover a range of potential head sizes, however the reduction in performance of the abrasion below 2 mm travel reduces the effective range. The methodology developed could be adapted to create new helmets of varying sizes, even based on a different reference model. Custom designed spring potentiometers with greater travel could also be included to increase the effective range of a single helmet design.

There are other issues with the helmet in its current form which must be addressed before attempting clinical measurements. Considerable design effort was taken to minimise the material in all 3D printed components, and incorporate low mass options for the servo motor and linear sensor, however, the overall mass of the helmet was approximately 1.8 kg inclusive of all wiring and connectors, which is close the maximum recommended for adults [48]. Mass reduction could be achieved through optimisation of the geometry and printing of the helmet models. Wiring to the sensors and servos could also be replaced by flexible printed circuit boards and integrated into the helmet body. The electrodes are removable but a material with a higher glass temperature is necessary for the 3D printed components to withstand the autoclaving process. The performance of the helmet should be tested in an environment replicating the mechanical vibration experienced by a patient in an ambulance, particularly the patient movement which can reach 3 g or greater in non-emergency transports [49]. While motion artefacts were not experienced in these experiments, further compensation may be required in ambulance conditions.

The electrode alignment was not normal to the surface across the entire scalp as a consequence of variations in the morphology on the reference model and the subject. Variations such as these are inevitable with a rigid body design and may lead to poor contact and undesirably high pressures. The gel on the electrode goes some way to improving contact with electrode misalignment, as shown in the linear array experiments Figure 12. This could be further mitigated with a ball joint type arrangement at the electrode tip, or by introducing a compliant element to the electrode shaft. The 10 mm diameter was chosen to match conventional EEG electrodes; however smaller electrodes may allow easier access to the scalp through thicker hair. Therefore, further investigations are required to find the optimum electrode geometry to improve access to the scalp with the highest possible contact area.

### 5.3. Technical Considerations and Recommendations for Future Work

Errors in the custom impedance measurement circuit were the likely cause of the sharp changes observed in the long-term recording Figure 12, where Elefix paste or Ag/AgCl electrodes reduced the capacitance and thus minimised the phase induced errors. These errors could be avoided by using a controller with a rail-to-rail DAC and full DSP capabilities. Using a dedicated impedance measurement IC, such as the AD5339 from Analogue Devices, which has been adapted for use in low frequency EIT by [50], would provide more accurate impedance measurement and reduce the computation necessary on the main controller itself. Delegating the impedance measurement to an IC would reduce the size of the central controller such that it could fit onto the helmet itself. The application time of the electrodes could be reduced further through implementation of separate controllers for subsets of electrodes, with multiple impedance measurement circuits with distinct carrier frequencies, this would reduce the memory and computation requirements for the controller, further aiding the miniaturisation necessary to incorporate the controller into the helmet.

## 6. Conclusions

Experiments were performed to quantify the mechanical specifications necessary for adequate skin abrasion performed by a self-abrading electrode. The results demonstrated that so long as the torque applied was sufficient to overcome friction, the effective reduction of contact impedance was dependent predominantly on the total angle rotated for a wide range of applied pressures. This placed less stringent requirements on the subsequent design of a self-contained unit than previously described in the literature. A simplified self-abrading electrode unit with servo motor and was able to replicate the performance of the large scale prototype and achieve the target of <5 kΩ contact impedance within 10 s. A prototype 32 channel electrode bearing helmet was constructed, which demonstrated the principle of simultaneous abrasion and electrode localisation successfully. While a considerable amount of work is required before it is ready for clinical experiments, this study lays the foundation for future research and development. 

## Figures and Tables

**Figure 1 sensors-20-07058-f001:**
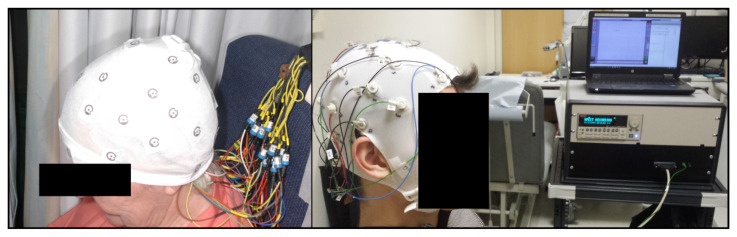
EIT data collection of acute stroke [10,28].

**Figure 2 sensors-20-07058-f002:**
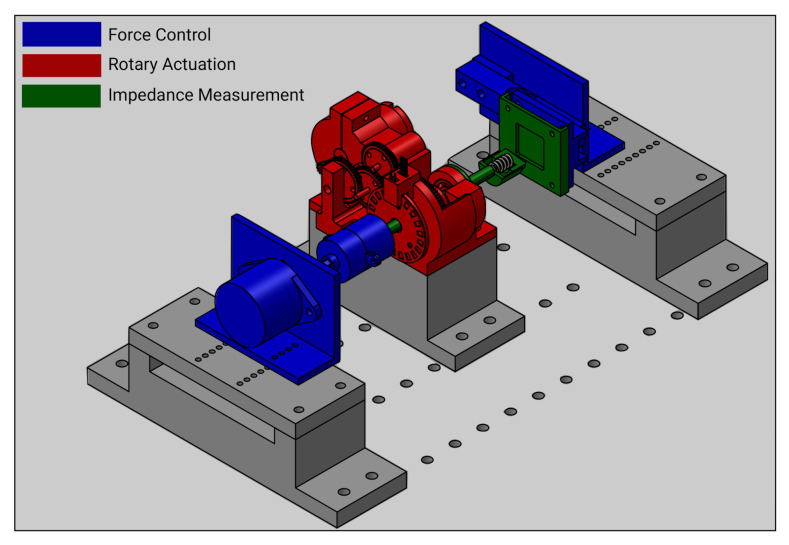
Test rig for abrasion characterisation, with independent force and rotary actuation.

**Figure 3 sensors-20-07058-f003:**
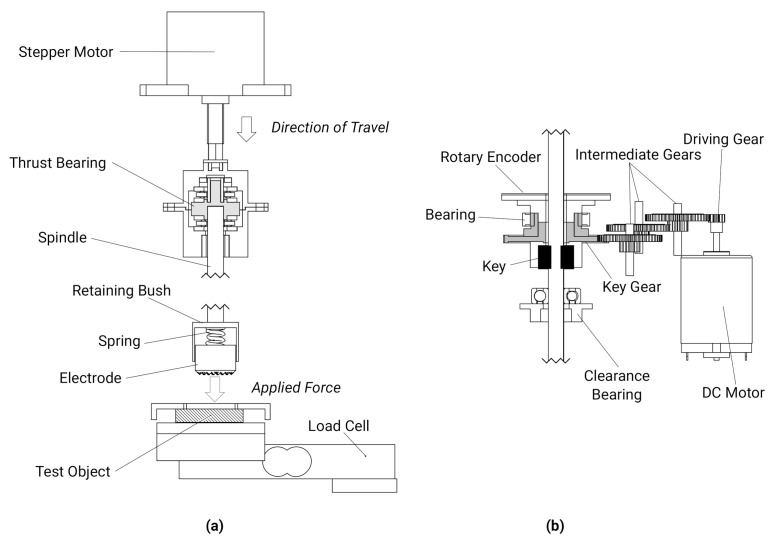
Mechanical components of abrasion characterisation test rig (**a**) applied force control via linear stepper motor, compliant element and load cell (**b**) rotation speed and position control via DC motor, transmission system and rotary encoder.

**Figure 4 sensors-20-07058-f004:**
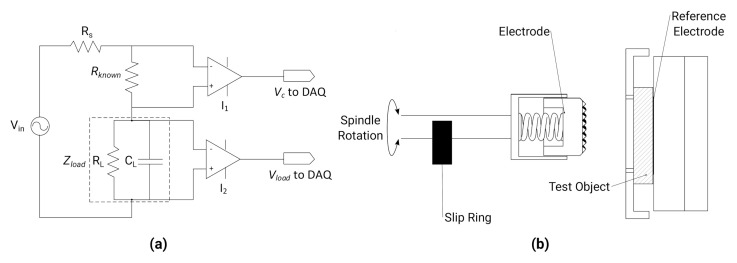
Impedance measurement (**a**) circuit diagram (**b**) electrical connections to reference and actuated electrode via slip ring.

**Figure 5 sensors-20-07058-f005:**
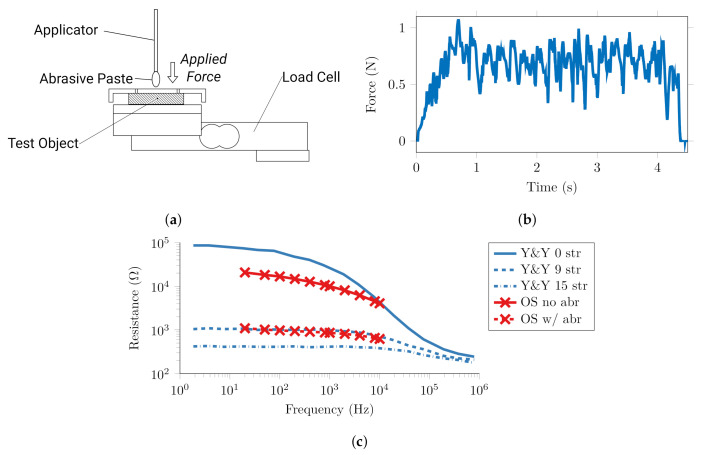
Characterisation of conventional manual abrasion via applicator and abrasive paste (**a**) experimental setup (**b**) typical force profile (**c**) comparison of orange skin (OS) impedance before and after abrasion, and impedance of human skin with 0, 9, and 15 strippings from Yamamoto and Yamamoto [41] (Y&Y).

**Figure 6 sensors-20-07058-f006:**
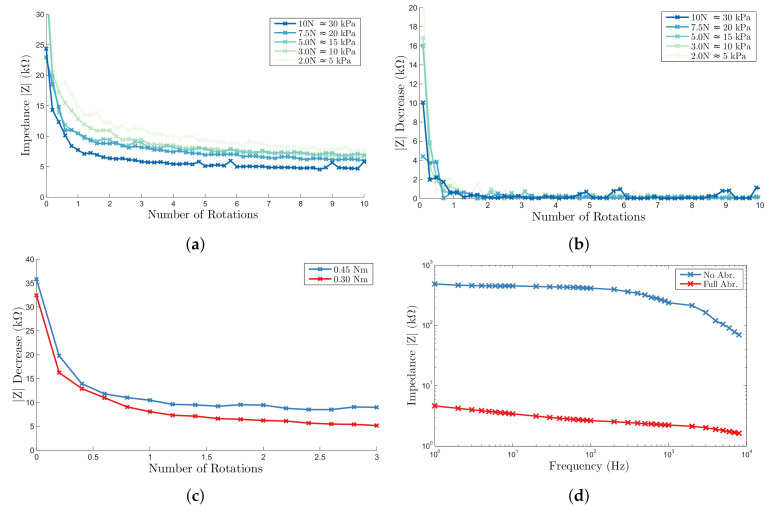
Automatic abrasion results (**a**) impedance decrease with electrode rotations across force range (**b**) impedance decrease per sixth of a rotation (**c**) comparison of impedance decrease for minimum and maximum torque (**d**) impedance spectra of orange skin test samples before and after abrasion.

**Figure 7 sensors-20-07058-f007:**
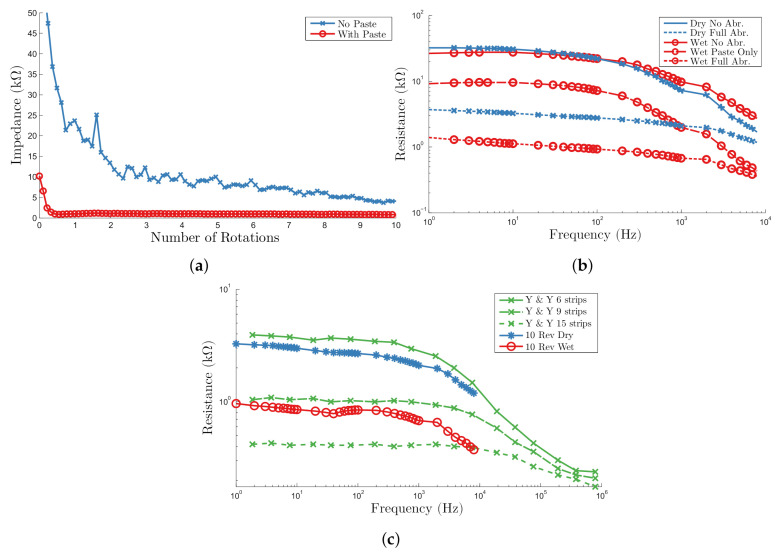
Human forearm results (**a**) impedance decrease during electrode rotation, with and without abrasive conductive paste (**b**) impedance spectra before (solid) and after (dashed) abrasion with and without paste and with the addition of paste only (dotted) (**c**) comparison of corrected resistance spectra after wet and dry abrasion with the spectra after 9 and 15 strips fromYamamoto and Yamamoto [41] (Y&Y).

**Figure 8 sensors-20-07058-f008:**
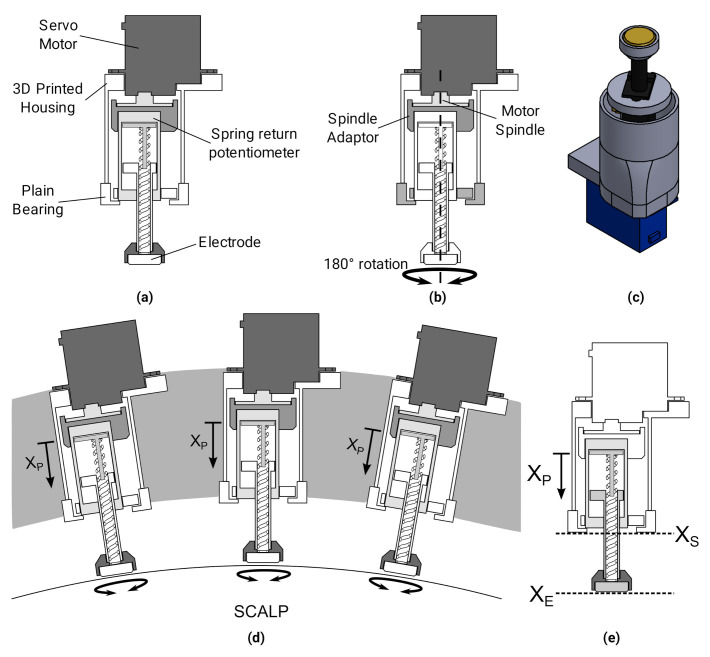
Self-Abrading Electrode Unit Design (**a**) overview of components (**b**) concentric rotary actuation via servo motor with plain bearing, (**c**) complete unit (**d**) units placed in rigid helmet at known positions (**e**) feedback of electrode position $X_{E}$ through measurement of displacement $X_{P}$.

**Figure 9 sensors-20-07058-f009:**
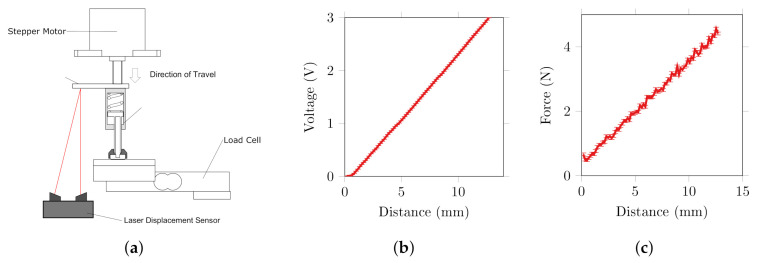
Spring return position sensor characterisation (**a**) measurement setup (**b**) voltage across compression mean ± std (**c**) force measured during compression mean ± std.

**Figure 10 sensors-20-07058-f010:**
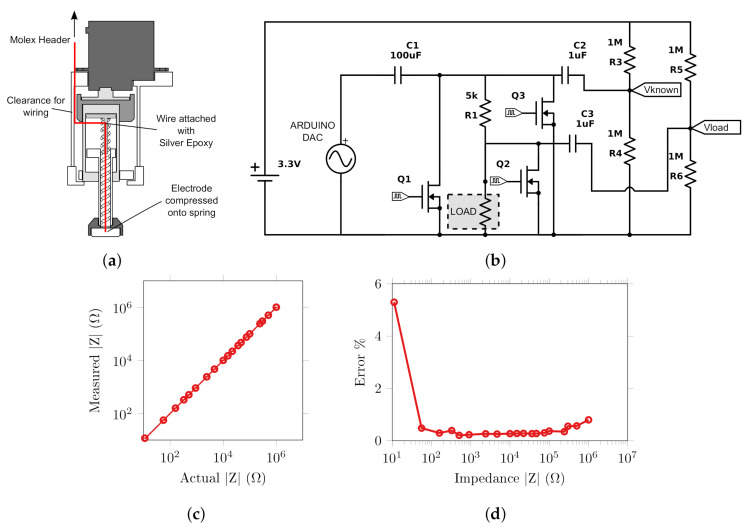
Single Unit Impedance Measurement (**a**) wiring path to electrode (**b**) circuit diagram (**c**) impedance measured across expected measurement range compared to ideal (**d**) percentage error.

**Figure 11 sensors-20-07058-f011:**
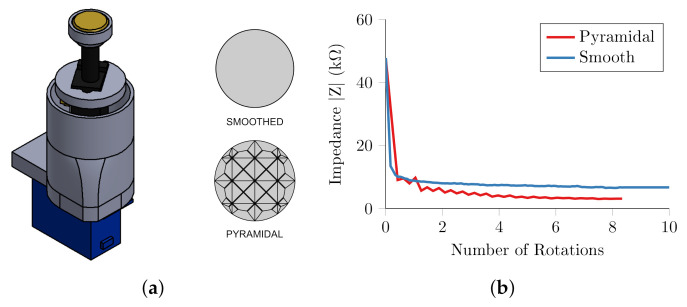
Single Unit Abrasion Experiments (**a**) unit with pyramidal abrasive pattern on smooth (**b**) abrasion in orange skin test rig for both patterns using abrasive paste.

**Figure 12 sensors-20-07058-f012:**
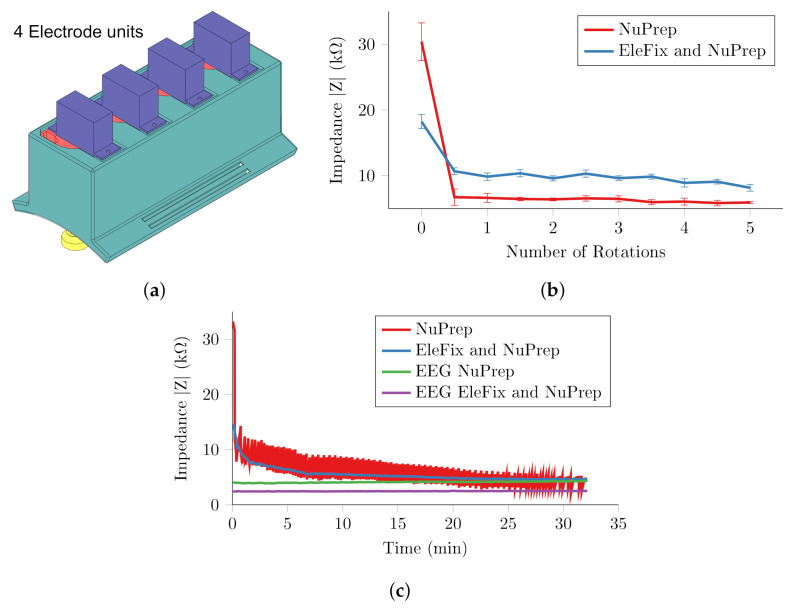
Linear array of four units (**a**) electrode housing (**b**) abrasion with application of abrasive Nuprep paste and with EleFix EEG gel and Nuprep paste (**c**) long term recording with two actuated electrodes and two conventional EEG electrodes.

**Figure 13 sensors-20-07058-f013:**
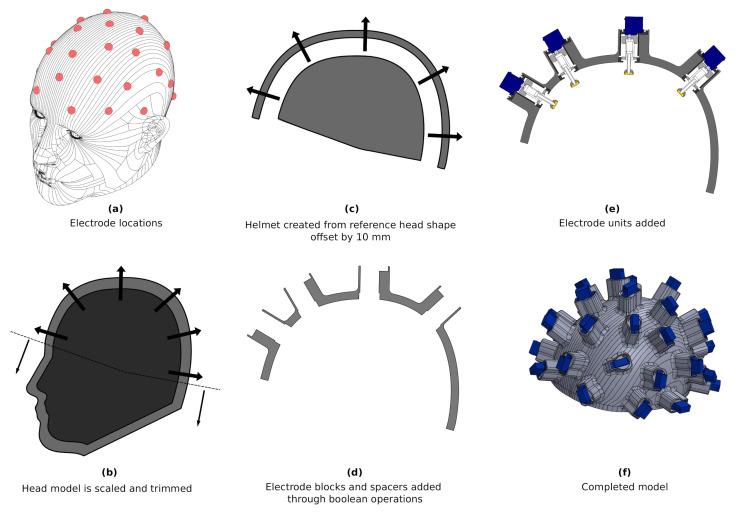
Design of electrode bearing helmet (**a**) 32 electrode locations (**b**) the scalp model is scaled to match the nominal head dimensions [44] and model trimmed below in inion-nasion line, (**c**) helmet frame constructed from 10 mm offset of nominal scalp surface, (**d**) electrode blocks are added to model and spacers then removed to create housings for each electrode unit aligned to the electrode positions, (**e**) electrode units added to complete model (**f**) final helmet model.

**Figure 14 sensors-20-07058-f014:**
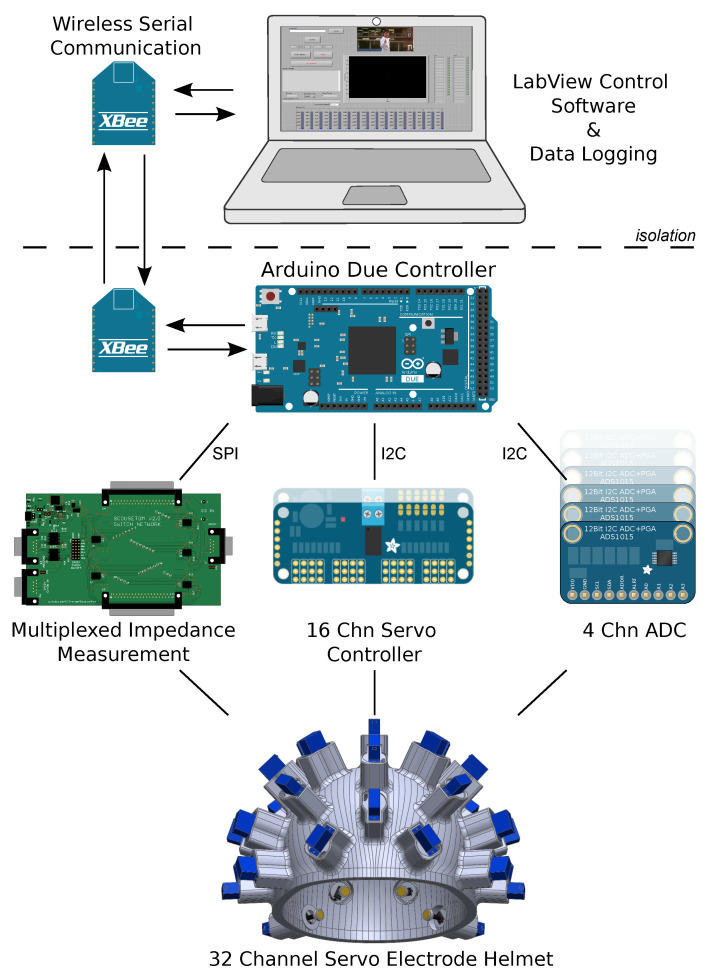
Overview of helmet controller. Labview software controlling Arduino Due through wireless serial communication. Ardruino controls electrode rotation, impedance and position measurements.

**Figure 15 sensors-20-07058-f015:**
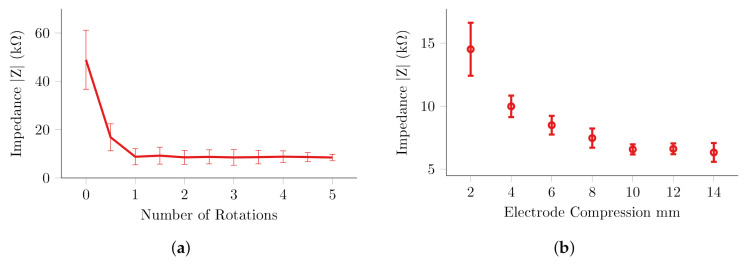
32 Channel Electrode Helmet (**a**) abrasion results for all electrodes (**b**) minimum impedance achieved across range of compressions.

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
