# Peer review of "Self-Abrading Servo Electrode Helmet for Electrical Impedance Tomography"

_sensors, 2020, doi:10.3390/s20247058_

Round 1
Reviewer 1 Report
RE:
Manuscript ID: sensors-1022160
Title: Self-Abrading Servo Electrode Helmet for Electrical Impedance
Tomography
Authors: James Avery *, Brett Packham, Hwan Koo, Ben Hanson, David Holder
Submitted to section: Biomedical Sensors
I would like to commend the authors on an interesting, and well-presented piece of work. I am overall impressed with the paper and have only some minor comments I would like addressed.
(1) The authors in the abstract mention the high sensitivity of EIT to errors in scalp electrode positioning and poor electrode-skin contact. Then they say this paper will overcome these limitations. However:
- I can’t see how this work addresses the positioning problem. I can see reference to the distance the electrode is ‘pushed in’ to aid in the impedance issue, but not errors in the intended position on the scalp?;
- While electrode contact impedance is an error source for EIT, the allowable range of contact impedance errors allowed seems to be wide judging by Malone et al. 2014 (10.1088/0967-3334/35/6/1051), McDermott et al. 2019 (10.1109/JBHI.2019.2960862) - up to ± 50%. So how ’stringent’ are the requirements for good impedance control?
(2) In Section 2.1 the authors refer to an “orange skin test”. I take this to mean the skin of an orange was used (i.e. the fruit). This was a little confusing. I see it is explained later in the text, and justified as a proxy for human skin but a brief explanation on first use would be helpful.
(3) Again, in Section 2.1 some information on the electrode size/ shape would be useful to put these experiments into context. Again, this information is provided later in the text. Are the electrodes used later the same in nature to the ones used in these earlier experiments?
(4) Fig. 5- What is Y&Y?. And immediately under Fig. 5 in the text, what is the contact area giving 0.62 N / 24.8 kPa.
(5) On the electrode contact area query – did any investigation look at whether there was uniform contact between the electrodes and the surface. Would non-uniform contact and so non-unform impedance across the electrode surface affect EIT data collection?
(6) Proof of principle on human skin: Is the skin of the forearm similar enough to the scalp?
Especially near the elbow where the skin is quite thick. Also have you considered the problem of hair on the scalp? Was hair a problem in the case where the helmet was used on a volunteer?
(7) Was there any investigation done into different electrode size and shapes? I see a smooth versus pyramidal analysis but anything beyond this?
(8) Was there experiments done where EIT data was taken at the same time as using the self-abrading electrodes or helmet?
(9) Is there any safety concerns about using these electrodes on skin? For example, friction burns or skin irritation.
(10) Were there any issues with movement of the electrodes on the scalp or skin in the human experiments? (I see mention of position but again I think this refers to depth of electrode position as opposed to movement across the skin?).

Author Response
Please see attached document for properly formatted response.
Reviewer 1
I would like to commend the authors on an interesting, and well-presented piece of work. I am overall impressed with the paper and have only some minor comments I would like addressed.
We greatly appreciate the supportive words and the time taken to review our paper in such detail. We agree these issues required further clarity and the manuscript has improved as a result of your input.
(1) The authors in the abstract mention the high sensitivity of EIT to errors in scalp electrode positioning and poor electrode-skin contact. Then they say this paper will overcome these limitations. However:
- I can’t see how this work addresses the positioning problem. I can see reference to the distance the electrode is ‘pushed in’ to aid in the impedance issue, but not errors in the intended position on the scalp?
To a certain extent the EIT algorithms are more robust to unintended but known positions than to unknown location errors which this helmet addresses my measuring the displacement along a known vector from the surface of the rigid helmet. The helmet aids with alignment by fixing the displacement along vectors which are based on the desired EEG 10-20 coordinates. However, should the subjects anatomy deviate substantially from the reference model then this assumption does not hold. We have expanded the discussion to address this point as follows:
“Further experiments are necessary to assess the accuracy of positioning the electrodes in the target locations on varying subject anatomy. Whilst EIT algorithms have been shown to be sensitive to random electrode localisation errors of approximately 1 mm [15,19], the sensitivity to errors arising from misalignment of the helmet, which can be approximated as rotations around each anatomical axis, have yet to be tested. The BFSD-EIT algorithm has been shown to be robust to errors of up to 30mm if symmetry is maintained [21], which may be the case for rotations of the helmet around the lateral axis.”
- While electrode contact impedance is an error source for EIT, the allowable range of contact impedance errors allowed seems to be wide judging by Malone et al. 2014 (10.1088/0967-3334/35/6/1051), McDermott et al. 2019 (10.1109/JBHI.2019.2960862) - up to ± 50%. So how ’stringent’ are the requirements for good impedance control?
We agree this is an important point. Simulations of contact impedance vary the parameter used for the complete electrode model, which does not take into account the instrumentation related errors caused by the increased contact impedance. Although some work has been done to include these errors within a joint SPICE/EIDORS framework, they are not commonly used or adapted for multifrequency EIT Dimas et al 2020 (10.3390/technologies8010013). It is these instrumentation errors which define the requirements for good impedance control. We have added clarification to section 1.1.3:
“Whilst EIT algorithms are robust to a wide range of contact impedances [15,21] high contact impedance still results in adverse errors due to instrumentation nonidealities. This is primarily due to the current injection, which is more sensitive to contact impedance mismatch …”
[15] Malone et al. 2014 (10.1088/0967-3334/35/6/1051)
[19] Jehl et al. 2015 (10.1088/0967-3334/36/12/2423)
[21] McDermott et al. 2019 (10.1109/JBHI.2019.2960862)
(2) In Section 2.1 the authors refer to an “orange skin test”. I take this to mean the skin of an orange was used (i.e. the fruit). This was a little confusing. I see it is explained later in the text, and justified as a proxy for human skin but a brief explanation on first use would be helpful.
Thank you for bringing this to our attention, we agree this should be explained earlier in the text. We have added the following to the introduction paragraph in section 2.1:
“To simulate the impedance and mechanical properties of human skin, test objects were created from sheets of navel orange skin (Citrus sinensis). The suitability of which was tested in experiments comparing the impedance spectra of human skin before and after abrasion.”
(3) Again, in Section 2.1 some information on the electrode size/ shape would be useful to put these experiments into context. Again, this information is provided later in the text. Are the electrodes used later the same in nature to the ones used in these earlier experiments?
The same brass actuated and silver reference electrodes were used in all experiments in Section 2. We have clarified the introduction paragraph in section 2:
“… between the 10 mm brass actuated electrode, and a 50x70mm silver reference electrode behind …”
(4) Fig. 5- What is Y&Y?. And immediately under Fig. 5 in the text, what is the contact area giving 0.62 N / 24.8 kPa.
Thank you for pointing out this omission. Y&Y refers to the reference [41] (Yamamoto, T.; Yamamoto, Y. Electrical properties of the epidermal stratum corneum. Medical & Biological Engineering 1976, 14, 151–158. doi:10.1007/BF02478741), which should have been referred to by name in the caption and Figure 5c and 7c. This has been corrected
We agree that the surface area required greater clarification, the 25mm2 mentioned in the previous paragraph is in fact the contact surface area. i.e. the area of tip of the applicator which is in contact with the tissue during abrasion. We have clarified the following sentences in section 2.2:
“… using NuPrep gel and an applicator with a surface area of approximately 25 mm2 in contact with the tissue, and the contact impedance decrease…”
“…was 0.62 N ±0.09, which equates to a pressure of 24.8 kPa over the 25 mm2 surface area.”
(5) On the electrode contact area query – did any investigation look at whether there was uniform contact between the electrodes and the surface. Would non-uniform contact and so non-unform impedance across the electrode surface affect EIT data collection?
The study did not specifically check for contact uniformity as it was difficult to get line of sight during use. However, the experiments on the large scale test rig showed a uniform abrasion pattern on the orange skin when using the abrasive electrode with the paste. Conventional EIT forward solvers implement the boundary conditions through the Complete Electrode Model with a uniform contact impedance, thus if the abrasion were significantly non-uniform this may lead to further modelling errors as the current density would not be modelled correctly.
(6) Proof of principle on human skin: Is the skin of the forearm similar enough to the scalp?
Especially near the elbow where the skin is quite thick. Also have you considered the problem of hair on the scalp? Was hair a problem in the case where the helmet was used on a volunteer?
Thank you for raising this important point which warrants discussion. In the literature, skin is typically given a single value across the body https://itis.swiss/virtual-population/tissue-properties/database/low-frequency-conductivity/ and most values have been collected on the forearm. Histology shows that the outer stratum corneum layers are comparatively similar between the areas of the body used in this study https://link.springer.com/article/10.1007/s004030050453 12 ± 2 for scalp, 16 ± 4 for foream flexor. The elbow was used only for the reference electrode. The shin, although likely thicker stratum corneum (16 ± 4 for thigh), it was chosen for the array as it has similar elastic properties to the scalp i.e. a thin layer of skin directly above bone, and was easier to align compared to the forearm. We have added the following to section 2.4:
“…automated abrasion was performed on the forearm of a human subject. The forearm was chosen as it has a similar a thickness of stratum corneum to the scalp [42].”
The volunteer in these experiments had comparatively short hair, and the rotation action parted the hair to a certain extent during use. However, this may not be the case with subjects with longer or tightly braided hair and should be included in future work. This has been added to the discussion along with the response to the next comment.
[42] Ya-Xian et al. https://doi.org/10.1007/s004030050453
(7) Was there any investigation done into different electrode size and shapes? I see a smooth versus pyramidal analysis but anything beyond this?
In this study we replicated the typical shape and size of the EEG electrodes as these are what are used in previous EIT studies. Generally, this surface area is what is required to attain desirable contact impedance values. However, the trade-off between contact area and ability to reach the scalp through hair is one with feel should be investigated further. We have added the following text to the discussion at the end of section 5.2:
“The 10 mm diameter was chosen to match conventional EEG electrodes; however smaller electrodes may allow easier access to the scalp through thicker hair. Therefore, further investigations are required to find the optimum electrode geometry to improve access to the scalp with the highest possible contact area.”
Other abrasive patterns using spiral or knurled patterns were prototyped but were not considered worth the extra time and cost in manufacturing given the results of the smoothed surface with paste.
(8) Was there experiments done where EIT data was taken at the same time as using the self-abrading electrodes or helmet?
Preliminary testing was performed with using the UCL ScouseTom system to verify the impedance measurement of the helmet did not interfere with the current injection of the EIT system. With the exception of the switching artefact the data was uncorrupted. However, it was not possible with the system in its current form to synchronise the timing of the two, to avoid the switching artefacts. This is feasible with an updated version of the ScouseTom and firmware this should be possible and would enable some interesting experiments in the future.
(9) Is there any safety concerns about using these electrodes on skin? For example, friction burns or skin irritation.
Qualitatively the pressures and rotations from the electrodes (especially with the abrasive paste) cause less discomfort than that experienced in conventional manual abrasion by a trained EEG technician. The stainless-steel electrodes, whilst not ideal from an impedance standpoint, are commonly used in other medical applications and no not have a particular safety concerns. We feel the largest safety concern with the design, as mentioned in the discussion, the weight of the helmet requires further reduction, as it is too heavy to wear continuously for monitoring type applications. However, we are clear that any future iterations of the design should undergo more rigorous safety evaluations before any clinical use.
(10) Were there any issues with movement of the electrodes on the scalp or skin in the human experiments? (I see mention of position but again I think this refers to depth of electrode position as opposed to movement across the skin?).
No issues were experienced in these experiments as the array of 4 electrodes were held in place sufficiently by the bandage, and the helmet was stable during the abrasion process. However, the issue of movement artefact during an ambulance journey were patient movement was likely be greater than that experienced in these experiments.
“The performance of the helmet should be tested in an environment replicating the mechanical vibration experienced by a patient in an ambulance, particularly the patient movement which can reach 3 g or greater in non-emergency transports [49]. Whilst motion artefacts were not experienced in these experiments, further compensation may be required in ambulance conditions.”
[49] Shah et al J. Perinat. Med. 36, 87–92. https://doi.org/10.1515/JPM.2008.009

Reviewer 2 Report
The authors have presented a novel electrode helmet, bearing 32 independently controlled self-abrading electrodes. The experiment results demonstrated the potential of this approach to rapidly apply electrodes in an acute setting, removing a significant barrier for imaging acute stroke with EIT. So, I would recommend this paper for publication.
It is recommended that Figure 6(a) (b) and Figure 7(b) (c) be modified, such as Color and Legend of curves.
Author Response
Please see attachment for formatted response.
The authors have presented a novel electrode helmet, bearing 32 independently controlled self-abrading electrodes. The experiment results demonstrated the potential of this approach to rapidly apply electrodes in an acute setting, removing a significant barrier for imaging acute stroke with EIT. So, I would recommend this paper for publication.
It is recommended that Figure 6(a) (b) and Figure 7(b) (c) be modified, such as Color and Legend of curves.
Thank you for reading our paper. We agree the figures could be improved. We have changed the colour scheme for 6(a)(b) and clarified the text in the legends and caption in 7.

Reviewer 3 Report
The authors present a electrode helmet featuring control of electrode contact impedance through pressure/abrasion. The paper is well written.
All my comments are minor and are suggestions.
My main comment regards the measure of impedance. The section 2.1.3 and subsequent figures do not clarify how the measurements are demodulated, and whether it is the module or the complex impedance which is shown as impedance in subsequent results. A potential sequence to this study could be analysing the complex impedance and interpret the changes according to standard electrode models.
Another comment is the discussion of the angle between the electrode and the skin. This is mentioned in the discussion, but it may be beneficial to mention it shortly earlier in introductory sections.
Again, congratulations to the authors for the good quality of the manuscript, which I encourage being published in Sensors.
Author Response
Properly formatted response is attached.
